# Preparation of Porous Activated Carbons for High Performance Supercapacitors from Taixi Anthracite by Multi-Stage Activation

**DOI:** 10.3390/molecules24193588

**Published:** 2019-10-05

**Authors:** Xiao-Ming Yue, Zhao-Yang An, Mei Ye, Zi-Jing Liu, Cui-Cui Xiao, Yong Huang, Yu-Jia Han, Shuang-Quan Zhang, Jun-Sheng Zhu

**Affiliations:** 1Key Laboratory of Coal Processing and Efficient Utilization (Ministry of Education) and School of Chemical Engineering and Technology, China University of Mining and Technology, Xuzhou 221116, China; azy15050837569@163.com (Z.-Y.A.); ye19851623526@163.com (M.Y.); zj18361272159@163.com (Z.-J.L.); 19851625722@163.com (C.-C.X.); 18361227609@163.com (Y.-J.H.); zhujschina@163.com (J.-S.Z.); 2College of Materials Science and Engineering, Nanjing Forestry University, Nanjing 210037, China; huangyong@njfu.edu.cn

**Keywords:** supercapacitor, multi-stage activation, coal-based electrodes, activated carbon, electrochemical performance

## Abstract

Coal-based porous materials for supercapacitors were successfully prepared using Taixi anthracite (TXA) by multi-stage activation. The characterization and electrochemical tests of activated carbons (ACs) prepared in different stages demonstrated that the AC from the third-stage activation (AC_III_) shows good porous structures and excellent electrochemical performances. AC_III_ exhibited a fine specific capacitance of 199 F g^−1^ at a current density of 1 A g^−1^ in the three-electrode system, with 6 mol L^−1^ KOH as the electrolyte. The specific capacitance of AC_III_ remained 190 F g^−1^ even despite increasing the current density to 5 A g^−1^, indicating a good rate of electrochemical performance. Moreover, its specific capacitance remained at 98.1% of the initial value after 5000 galvanostatic charge-discharge (GCD) cycle tests at a current density of 1 A g^−1^, suggesting that the AC_III_ has excellent cycle performance as electrode materials for supercapacitors. This study provides a promising approach for fabricating high performance electrode materials from high-rank coals, which could facilitate efficient and clean utilization of high-rank coals.

## 1. Introduction

With the depletion of fossil energy, the storage and conversion of renewable energy has become an urgent problem in the world today [1]. As a kind of high-efficiency energy storage device, supercapacitors have attracted people’s attention due to their characteristics of fast charge-discharge rate, high coulombic efficiency, excellent rate performance, and long cycle life [2,3,4,5]. Electrode materials are the key factors determining the performance of supercapacitors. At present, they are mainly divided into three categories: carbon materials, transition metal oxides, and conductive polymers [6,7]. Carbon materials mainly include activated carbon (AC) [8,9], carbon fiber [10,11], carbon nanotubes [12,13], carbon nanosheets [14,15], graphene [16,17,18], etc. 

There are abundant coal resources in China, and coal will remain a major source of energy for quite a long time in China. It is very significant and meaningful to research how to use coal efficiently and cleanly, and convert cheap coal into usable materials. As is well-known, coal has become the most commonly used precursor for production of AC because of its abundant resources, low cost, and high carbon content [19], and coal-based porous carbon has been widely researched for supercapacitor electrodes. Dong et al. [20] reported interconnected porous carbon nanosheet/nickel foam composites which were obtained from coal tar pitch coupled with KOH activation. Qin et al. [21] synthesized the interconnected porous carbons from coal tar pitch and microcrystalline cellulose for high-performance supercapacitors. Wang et al. [22] reported a porous carbon with the 1851 m^2^ g^−1^ specific surface area prepared from Xinjiang anthracite through chemical activation with ZnCl_2_, where the specific capacitance reaches 178 F g^−1^ at 1 A g^−1^. The chemical activators such as KOH were usually added to coal in solid state to prepare coal-based electrode materials in most papers [23,24]. Although this method could produce large surface areas AC, it led to a large amount of activator (activator and coal with mass ratios of 3:1 to 5:1, commonly) and serious corrosion of equipment because the utilization of activator is low, more importantly, the surface areas of the AC has not been effectively utilized through the energy storage process. Therefore, it is necessary to find a low-cost, simple, and environmentally friendly method to produce high-performance supercapacitors. In this work, impregnation method was used to prepare AC with Taixi anthracite (TXA) as a precursor. First, the columnar coal strips were carbonized and physical activated with CO_2_, then chemically activated after immersion with KOH solution, next, re-impregnated with KOH solution, and activated again. CO_2_ activation can make the material have a certain pore structure, which is beneficial to improving the effect of the impregnation method. It is found that the electrochemical performance of the ACs can be effectively improved by increasing the number of impregnation and activation processes. In addition, it is worth noting that the mass ratio of the activator (KOH) to carbon is only about 1:3 in both chemical activations that followed. The method of impregnation can reduce the KOH application amount so that the risk of corrosion of equipment can be reduced too, and the remaining KOH can be recycled. 

## 2. Results and Discussion 

### 2.1. Microstructure and Composition

The SEM images of TXA and the ACs obtained in each stage are shown in Figure 1. As shown in Figure 1a, TXA has a dense surface and almost no obvious pores, which is attributed to the high degree of coalification of anthracite, which has relatively low porosity. As illustrated in Figure 1b, after the first stage of physical activation, some observable pores appear on the surface of the AC_I_ and there are some irregular fragments on the surface and in the crack of the AC_I_. As Figure 1c shows, after the second stage of chemical activation, more pores (the large pores we can see on the SEM image) appear on the surface of the AC_II_ than AC_I_. This means the number of macropores of the AC_II_ is increased, which can shorten the distance of electrolyte ions diffusing into the micropores in supercapacitor, thereby improving the electrochemical properties of electrode material [25,26]. Irregular fragments of AC_II_ are reduced due to washing with HCl solution and deionized water. AC_II_ was used as the raw material to repeat the impregnation with the KOH activator and activation process to obtain AC_III_. As illustrated in Figure 1d, compared with the SEM images of TXA, AC_I_, and AC_II_, the surface of AC_III_ is more abundantly porous, and the AC_III_ appears to be loose and porous. It is speculated that AC_III_ could have a better electrochemical performance as electrode materials. Furthermore, the TEM images of AC_III_ (Figure 1e,f) show that its amorphous structure has a large number of pores [21].

The XRD patterns of TXA, AC_I_, AC_II_, and AC_III_ are presented in Figure 2. Due to the high degree of coalification, the sharp peak of 26° and the weak peaks of 43° are shown in the XRD of TXA, corresponding to the reflection of the (002) plane and (100) plane of the aromatic layer, respectively, indicating the presence of a microcrystalline graphitized structure [20,27,28]. Compared with TXA, the peak of AC_I_ at 2θ = 26° becomes relatively gentle, indicating that the degree of graphitization of AC decreases after physical activation. In the XRD patterns of AC_II_ and AC_III_, the (002) peak becomes weaker due to the internal erosion process of KOH activation [24,29], and the accumulation structure of the aromatic layer further changes to the amorphous structure, resulting in an increase in pores and facilitating the storage of charges. The small peak appeared at 44° of AC_III_ can be indexed to a superposition of the (101) reflections of the graphite structure [30]. This shows that AC_III_ after three activations has a special structure between the disordered amorphous carbon phase and the graphitic phase. 

The electrochemical properties of activated carbon are also affected by the functional groups on the surface. Therefore, FTIR tests were carried out on the samples. As shown in Figure 3, the absorption peaks at 2972 and 2920 cm^−1^ correspond to the stretching vibration of C-H [31], and the absorption peak at 1380 cm^−1^ corresponds to the in-plane bending vibration of C-H [32]. The absorption peaks at 845, 795, and 742 cm^−1^ conform to the out-of-plane bending vibration of C-H [33]. Moreover, the peak at about 1640 cm^−1^ is consistent with the stretching vibration of C=C, and the peak at 1610 cm^−1^ matches with the stretching vibration of the aromatic skeleton, indicating that there is a certain degree of graphitization structure in TXA, which is consistent with the analysis of XRD. It is found that after physical activation, the absorption peaks of ACFS at 1640 and 1610 cm^−1^ become weak, indicating that the activation process could reduce the graphitization degree of the sample, which is also confirmed in the XRD analysis. In addition, the absorbance for the aromatic skeleton stretching vibration in the FTIR spectrum is further weakening after chemical activation. All samples show a relatively wide band at 3430 cm^−1^, corresponding to the stretching vibration of hydroxyl [32,34]. TXA has a distinct band at 1040 cm^−1^, corresponding to the bending vibration of C-O [6], which gradually weakens in AC_I_, AC_II_, and AC_III_, indicating partial cleavage of C-O bonds during activation. 

Two distinct peaks appear in the XPS spectra of TXA and the samples at approximately 285 and 532 Ev, corresponding to C1s and O1s, as shown in Figure 4a. The C1s spectra of the samples are separated into four peaks by curve fitting, and are located at the binding energy of 284.8, 285.0, 286.4, and 288.9 Ev, corresponding to C-C, C-O, C=O, and O-C=O functional groups [26,35]. This also proves the presence of oxygen-containing functional groups on the surface of the samples. It has been reported that the wettability of the material in aqueous solution is improved due to the presence of oxygen-containing functional groups [36,37], which also contributes to the improvement of the specific capacitance of the electrode.

N_2_ adsorption-desorption analysis of AC_I_, AC_II_ and AC_III_ was performed, as shown in Figure 5a. According to the classification of International Union of Pure and Applied Chemistry (IUPAC), the N_2_ adsorption-desorption isotherms of Acs belong to the combination of Type I isotherm and Type IV isotherm, indicating that there are a large number of micropores and a certain number of mesopore in the three samples [26]. The AC_III_ shows a large adsorption capacity in the range of low relative pressure and a very obvious H4 hysteresis loop in the range of 0.4 to 1 relative pressure, proving a large number of mesoporous pores in AC_III_ [24]. As illustrated in Figure 5b, most of the pore diameters of AC_I_ are below 1 nm and only a few pores diameters of AC_Ⅰ_ range from 3 to 4 nm. The pore diameters of AC_II_ are bigger than AC_I_ in the range of 0.5–1 nm, revealing that the KOH exhibits a remarkable effect of hole-expanding during the first chemical activation. The pores of AC_II_ were further developed through the process of impregnation and chemical activation again to produce AC_III_, not only the number of the pores with diameters of 0.5–2.5 nm increased, but also a large number of mesopores appeared at a diameter of 3–4 nm. Chmiola et al. found that the pores with diameters of 0.6–1 nm can effectively increase the specific capacitance [38]. Specific capacitance can be increased by increasing the number of micropores with suitable pore diameter. In addition, increasing the number of mesopores can effectively increase the ion diffusion channel, reduce the diffusion resistance and improve the utilization of micropores, thereby facilitating the electrochemical performance of the material [39]. 

The pore structure parameters of the samples are listed in Table 1. Compared with AC_I_, the specific surface area (S_BET_) and micropore volume (V_mic_) of AC_II_ are reduced, but the mesopore volume (V_mes_) and average pore diameter (D_ap_) are increased, indicating that the carbon in the inner wall of the micropores reacts with KOH to enlarge the pore diameter through one-stage activation. The largest contribution to surface area is micropores. In other words, the more micropores of the same quality sample are available, the larger the specific surface area it has. So AC_II_ has more mesopores and macropores than AC_I_, and a larger average pore diameter than AC_I_, but the specific surface area of AC_II_ is smaller than AC_I_, because AC_I_ has more micropores than AC_II_. The S_BET_, total pore volume (V_t_), V_mic_ and V_mes_ of AC_III_ are obviously increased, proving that the AC which has undergone impregnation and chemical activation again exhibits more abundant pores, including the enlargement of small pores and generation of new pores. This is consistent with previous SEM analysis.

The pore structure of electrode materials has a great influence on electrochemical characteristics. The macropore, like a reservoir, is the place where ions are stored, and the mesopore is the channel for ions to be transported rapidly, moreover, the micropore provides the place for effective charge accumulation. The macropore, mesopore, and micropore are responsible for each other. Therefore, appropriate pore size distribution is conducive to improving the capacitance. As shown in Scheme 1, the raw coal itself has fewer pores. A large number of pores are formed in AC_I_ by one-stage activation, which is mainly microporous, but the few mesopores and macropores that appear in ions cannot move rapidly enough. In the second stage, KOH is impregnated into the pores of the AC_I_, while more pores with larger pore sizes were generated through the reaction of KOH with the carbon on the pore walls after second-stage activation. Therefore, the mesoporosity of AC_II_ increases. In the third stage, more KOH enter into the pores of AC_II_ through impregnation and react with the carbon on the pore walls of AC_II_. After that, part of the micropores of AC_II_ are reamed to form mesopores, and a large number of new, deeper and more developed micropores are generated around the mesopores. The specific surface area of the activated carbon increased from 591.3 m^2^ g^−1^ to 984.6 m^2^ g^−1^ after being activated twice.

The possible pathways for preparation strategy of AC and pore formation were proposed based on the above characterizations, as displayed in Scheme 1. Firstly, TXA was physically activated with CO_2_ to obtain AC_I_, which has a certain amount of micropores and a small amount of mesopores. Secondly, AC_I_ was impregnated in KOH solution, while K^+^ was attaching to the surface of AC_I_ and partly entering into the larger diameter pores. Then AC_I_ was chemically activated to produce AC_II_, and the pore size of AC_II_ increased obviously due to K^+^ etching. After the impregnation treatment for AC_II_, the pore structure was further developed in the subsequent chemical activation process because of the larger pore, leading to transferring more K^+^ into its interior structures. In addition, the use of the impregnation method can make the K^+^ relatively uniform into the pores of the ACs, which may reduce the local excessive etching and the pore collapse due to the uneven distribution of K^+^. The AC_III_ with more mescopores and high specific surface area was prepared through three-step activation.

### 2.2. Electrochemical Performance

The electrochemical properties of AC_I_, AC_II_, and AC_III_ were evaluated by galvanostatic charge-discharge (GCD), cyclic voltammetry (CV), and electrochemical impedance spectroscopy (EIS) tests with a 6 mol L^−1^ KOH aqueous solution as electrolyte solution in the three-electrode system. The GCD curves of the three samples at 1 A g^−1^ current density, as taken in Figure 6a, show that all curves are highly symmetric triangles, indicating that the electrode materials have typical double layer capacitor characteristics and good electrochemical reversibility [40,41]. Clearly, AC_Ⅲ_ shows the longest discharge time, indicating that it has the largest specific capacitance among the three samples under the same test conditions [42,43]. Besides, according to the Equation (1), the specific capacitances of AC_I_, AC_II_, and AC_III_ at 1 A g^−1^ are 81, 106, and 199 F g^−1^, respectively. Compared with AC_I_ and AC_II_, AC_III_ has a significant increase in specific capacitance, which is 2.45 times that of ACSF and 1.88 times that of AC_II_, proving the importance of re-impregnation and activation of activated carbon. In addition, the voltage drop of the three activated carbon samples at a current density of 1 A g^−1^ are shown in Figure 6b. The voltage drops of AC_I_, AC_II_, and AC_III_ are 0.0312, 0.0275, and 0.0148 V, respectively, and the values decrease in turn, indicating that the internal resistance of the electrode is reduced [21]. This may be related to the fact that multiple activation increases the pore size and reduces the diffusion resistance. Magnification performance is one of the key factors affecting the practical application of electrode materials. The specific capacitances of AC_I_, AC_II_, and AC_III_ at different current densities are shown in Figure 6c. It is clear that at the same current density, the specific capacitance of AC_III_ is much higher than AC_I_ and AC_II_, reaching 206 F g^−1^ at 0.5 A g^−1^. Even if the current density reaches 5 A g^−1^, the specific capacitance of AC_III_ is still 190 F g^−1^, showing excellent rate performance. This is due to the more reasonable pore size distribution of AC_III_, especially the increase of mesopores, which is conducive to the rapid transmission of electrolyte ions.

Figure 6d shows the CV curves for all samples at a scan rate of 10 mV s^−1^. All curves exhibit an approximately rectangular character, exhibiting excellent capacitive behavior, which illustrates that the capacitance of all samples is primarily derived from the electrical double-layer capacitance behavior [44,45,46]. Moreover, it is observed that the CV curve has a certain degree of distortion, which is due to the pseudo capacitance effect provided by the fast redox reaction of oxygen-containing functional groups [47]. It is well known that the integral area of the CV curve corresponds to the capacitance of the supercapacitor. Therefore, AC_III_ has the largest specific capacitance at the same scanning rate. Besides, the CV curve of AC_III_ at a scan rate of 5 to 100 mV s^−1^ is exhibited in Figure 6e. Even at relatively high scan rates, the CV curve of AC_III_ remains approximately rectangular, exhibiting good electrochemical behavior, due to the presence of a large number of mesopores facilitating the rapid transmission of electrolyte ions within the pores, which is in accordance with the analysis of GCD.

In order to make further evaluation of the ACs performance as electrode materials of supercapacitors, EIS analysis are performed. Figure 6f is the Nyquist plots of the samples, in which the illustration is an enlarged view of the high frequency region. Clearly, all curves show a diagonal line in the low frequency region and a semicircle in the high frequency region. In general, the slope of the linear portion of the Nyquist diagram in the low frequency region is related to the diffusion resistance caused by the diffusion/transmission of the electrolyte ions in the electrolyte and the electrode material, and the larger the slope, the closer the supercapacitor is to the ideal capacitor behavior [48,49,50]. Then the slope of AC_III_ is obviously greater than that of AC_I_ and AC_II_, indicating that AC_III_ has smaller diffusion resistance, which is consistent with the analysis of N_2_ adsorption and desorption test. The diameter of the semicircle in the high frequency region corresponds to the charge transfer resistance (R_ct_) [51,52]. It is clear that the R_ct_ of AC_I_, AC_II_, and AC_III_ in the illustration in Figure 6f increases sequentially, due to the decrease in the degree of graphitization of AC during the activation process, which is in accord with the XRD analysis. In addition, the solution resistance (R_S_) can be obtained from the Z’ axis intercept of the Nyquist plot [4], and the R_S_ values of all samples are very small, which also reflects the excellent electrochemical performance of the prepared materials. The cyclic stability is also an important index for evaluating the performance of supercapacitors. Therefore, the 5000 GCD cycle tests were executed for AC_III_ at 1 A g^−1^, as shown in Figure 7. It is worth noting that in the first 200 cycles, the specific capacitance of AC_III_ increases significantly, indicating that the AC electrodes undergo electrochemical activation in the incipient charge and discharge processes. And during electrochemical activation, the suitable charge-discharge cycles can promote the electrolyte ions to be completely inserted into the pores of AC, thus improving the availability of the surface area of the charge storage [53]. More impressively, the specific capacitance of the AC_III_ remained at 98.1% of the initial value after 5000 cycles, showing excellent cycle performance. Based on the above analysis, AC_III_ exhibits relatively high specific capacitance, excellent rate performance, and reliable cycle stability, which is more in line with practical application requirements.

## 3. Materials and Methods

### 3.1. Materials

Anthracite collected from Taixi, Ningxia province was taken as the precursor of ACs. Its proximate and ultimate analyses are shown in Table 2. Coal tar as the binder of columnar ACs was purchased from Jiangsu Weitian Chemical Group Company. KOH and hydrochloric acid used in the experiment were purchased from Sinopharm Chemical Reagent Co., Ltd. Acetylene black, anhydrous ethanol and potassium nitrate were purchased from Xiqiao Chemical Co., Ltd. All chemical reagents were of analytical grade and used directly without further purification.

### 3.2. Preparation of AC

#### 3.2.1. Pretreatment of TXA

TXA was crushed and ground to below 200 mesh. Then 100 g TXA thoroughly stirred by adding 2 g KNO_3_, 32.5 g coal tar and 5 ml deionized water. After that, it was pressed into cylindrical strips having a diameter of 2.8 mm by a plodder. Finally, the strips were naturally dried and broken until they were 1–2 cm long.

#### 3.2.2. The First-Stage Activation by Physical Activation

The treated strip was put into muffle furnace at room temperature and heated at 12 °C min^−1^ to 600 °C under nitrogen flow for carbonization. After carbonization, the sample was activated with CO_2_ as activator in a horizontal tube furnace with a heating rate of 12 °C min^−1^ from room temperature to 900 °C, and kept at 900 °C for 2 h. The sample was cooled to room temperature and then washed several times with 1 mol L^−1^ HCl solution, followed by washing with deionized water until the solution was neutral. Afterwards, the sample was dried at 80 °C for 24 h and denoted as the first-stage activated carbon (AC_I_).

#### 3.2.3. The Second-Stage Activation by Chemical Activation

AC_I_ was impregnated in 12 mol L^−1^ KOH solution and magnetically stirred for 24 h at room temperature. The sample was then filtered and dried at 80 °C for 24 h. In addition, the sample was weighed before and after immersion, and the corresponding mass ratio of alkali to carbon was about 1:3 by difference method. After that, the sample was placed in a tube furnace with N_2_ as a shielding gas and heated from room temperature to 800 °C at 12 °C min^−1^ for 1 h. Then the activated sample was cooled to room temperature and washed several times with 1 mol L^−1^ HCl solution, followed by washing with deionized water until the solution was neutral. The resulting columnar AC was then dried at 80 °C for 24 h and denoted as activated carbon from the second-stage activation (AC_II_).

#### 3.2.4. The Third-Stage Activation by Chemical Activation 

The procedure described in 2.2.3 for AC_II_ was repeated and the resulting AC was recorded as activated carbon for the third stage (AC_III_).

### 3.3. Characterizations

The microstructure of ACs was analyzed by scanning electron microscopy (SEM, Hitachi, Su8020) with a field emission scanning electron microanalyzer at 5 kV. Transmission electron microscope measurements were carried out on a microscope (TEM, FEI, Tecnai G2 F20) at 200 kV. X-ray diffraction (XRD) analysis was performed at 40 kV and 30 mA using a Bruker D8 Advance diffractometer with a Cu Ka X-ray source, the scan range was between 5 and 90°. Fourier transform infrared (FTIR) spectra of the samples were obtained by a Nicolet is 5 infrared spectrometer by using pressed KBr pellets. X-ray photoelectron spectroscopy (XPS) was implemented on an ESCALAB 250Xi (Thermo Fisher) instrument with monochromatized Al Kα probe beam. The energy scale was corrected with C1s peak at 284.8 eV as internal standard. The transmit power is 250 W and it was used for wide-range scanners and narrow scans over the ranges of 0–900 eV and 282–292 eV respectively, at a pass energy of 100 eV and 20 eV, respectively. The background was subtracted use a function of Shirley. The N_2_ adsorption-desorption isotherm was measured at 77 K using an Autosorb-1 type adsorbent manufactured by Quantachrome, and the specific surface areas (S_BET_) were calculated by the Brunauer-Emmett-Teller (BET) equation. The total pore volume (V_t_) of the sample was calculated from the relative pressure (P/P_0_) of 0.99. The pore size distribution was calculated by density functional theory (DFT). The average pore diameter (D_ap_) of the sample was calculated from Equation (1).
(1)$$D_{ap}=\frac{4V_{t}}{S_{BET}}$$

### 3.4. Electrochemical Measurement

Briefly, the AC we prepared was pulverized to pass through a 200-mesh sieve. 20 mg AC powder, 2.5 mg acetylene black, and 2.5 mg polytetrafluoroethylene were mixed into ethanol, followed by ultrasonic treatment for 5 min. Then the mixture was uniformly applied to long strips of foamed nickel, and dried in vacuum at 80 °C for 24 h. The active mass of the electrode was ca. 2 mg cm^−2^. All samples were electrochemically tested in a 6 mol/L KOH solution with nickel foam coated with ACs as working electrode, platinum sheet as the counter electrode, and Hg/HgO electrode as the reference electrode.

The GCD tests were performed at room temperature using a NEWARE BTS high precision battery detection system at a current density of 0.5–5 A g^−1^. Cyclic voltammetry (CV) tests were measured using a CHI66D electrochemical workstation (CH Instrument, Shanghai, China) at a voltage range of −1 to 0 V. Electrochemical impedance spectroscopy (EIS) measurements were conducted at an open circuit potential with an AC amplitude of 10 mV over a frequency range of 1 mHz to 100 kHz. The specific capacitance under three-electrode system was calculated according to Equation (2):(2)$$C=\frac{I\Delta t}{m\Delta V}$$
where *I* (A) is the discharge current; *m* (g) is the mass of the AC; Δ*t* (s) is the discharge time interval; Δ*V* (V) is the voltage difference during the corresponding discharge time.

## 4. Conclusions

In conclusion, a lot of micropores in AC_I_ were generated through carbonization and physical activation with CO_2_ as an activator. The volumes of micropores and mesopores in AC_III_ were increased significantly more than that in AC_I_ after two impregnations of KOH solution and two activations at 800 °C for 1 h. When using the ACs as active substances in the three-electrode system, the specific capacitance of AC_III_ was 206 F g^−1^ at 0.5 A g^−1^, much higher than AC_I_ and AC_II_. It shows higher specific capacitance, excellent rate performance, and good cycling stability of AC_III_. In addition, the impregnation method can reduce the dosage of KOH and reduce the corrosion of equipment during the activation process. The method adopted in this paper may pave the way for industrial production of carbon-based electrode materials for high high-performance supercapacitors.

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
