# Peer review of "Preparation of Porous Activated Carbons for High Performance Supercapacitors from Taixi Anthracite by Multi-Stage Activation"

_molecules, 2019, doi:10.3390/molecules24193588_

Round 1

Reviewer 1 Report

The manuscript entitled “Preparation of porous activated carbons for high performance supercapacitor from Taixi anthracite by multi-stage activation” describes the conversion of coal into activated carbon by a three-step process, with a thorough investigation of the resulting material at all three stages. The manuscript is detailed and provides strong explanation for the electrochemical properties. However, some improvements are needed to explain the observations, detailed comments follow below:

In the XRD of ACIII there is a sharp peak emerging a c.a. 44 degrees, this peak should be assigned and discussed. Why is the TXA sample not shown in XPS? The authors state in the first paragraph of results and discussion “after chemical activation, more pores appear on the surface of the ACII than ACI, and the number of macropores is significantly increased” yet this is represented by a decreased surface area in BET (of ACII compared to ACI). Can the authors explain why an increased pore size by chemical activation would present as a lower surface area? (Given Vmes increase for ACII compared to ACI which should be the highest surface area pores). The explanation in the text and schematic needs to be supported with higher resolution SEM images – the current figures (Fig 1) appears there should be an increasing surface area from ACI -> ACII. What were the errors of the measurement? Can the authors calculate the energy and power density from Fig 6? This would strengthen the paper. Experimental details under characterizations should be expanded, particularly for XPS where sample preparation and substrate play a large role in the result.

Author Response

Thank you for the valuable comments and suggestions on our manuscript. On behave of the authors, I significantly revised the manuscript, including text, table and figures, according the reviewers’ comments and all the revisions are marked in blue color in the revised text.

Please see the attachment for the response to Reviewer 1 Comments.

Reviewer 2 Report

This paper does not describe any new concept in the area of supercaps. it is however clearly written and can be considered for publication. I have two minor concerns regarding the paper. Quality of presentations of X-ray and Impedance data. In both cases based on the figures presented I am not fully convinced by the authors statements and assumptions based on results coming out from these data. Therefore quality of the presentation should be improved.

Author Response

Thank you for the valuable comments and suggestions from you on our manuscript. On behave of the authors, I significantly revised the manuscript, including text, table and figures, according the reviewers’ comments and all the revisions are marked in blue color in the revised text.

Please see the attachment for the Response to Reviewer 2 Comments.

Round 2

Reviewer 1 Report

The authors have addressed the most of the reviewers concerns regarding the manuscript. Minor comments:

TXA may be common in China, but any ore extracted from a mine can vary significantly. Further the readership of the article is intended to be global rather than simply chinese, therefore I think XPS merits inclusion in this case - to show the reader unfamiliar with TXA and without access to chinese textbooks the chemical change caused by the authors process.  The x-axis for cyclic voltammograms should have the units of V vs Hg/HgO.

Author Response

Thank you for your advise. The reply to the Review Report as follows:

Figure 4 has revised, and the XPS of TXA has added in the figure. The unit of x-axis for cyclic voltammograms have revised as of “Voltage (V) vs Hg/HgO”.